# Evolution of Neural Tangent Kernels under Benign and Adversarial Training

**Noel Loo, Ramin Hasani, Alexander Amini, Daniela Rus**
Computer Science and Artificial Intelligence Lab (CSAIL)
Massachusetts Institute of Technology (MIT)
`{loo, rhasani, amini, rus}` @mit.edu

## Abstract

Two key challenges facing modern deep learning are mitigating deep networks' vulnerability to adversarial attacks and understanding deep learning's generalization capabilities. Towards the first issue, many defense strategies have been developed, with the most common being Adversarial Training (AT). Towards the second challenge, one of the dominant theories that has emerged is the Neural Tangent Kernel (NTK) – a characterization of neural network behavior in the infinite-width limit. In this limit, the kernel is frozen, and the underlying feature map is fixed. In finite widths, however, there is evidence that feature learning happens at the earlier stages of the training (kernel learning) before a second phase where the kernel remains fixed (lazy training). While prior work has aimed at studying adversarial vulnerability through the lens of the frozen infinite-width NTK, there is no work that studies the adversarial robustness of the empirical/finite NTK during training. In this work, we perform an empirical study of the evolution of the empirical NTK under standard and adversarial training, aiming to disambiguate the effect of adversarial training on kernel learning and lazy training. We find under adversarial training, the empirical NTK rapidly converges to a different kernel (and feature map) than standard training. This new kernel provides adversarial robustness, even when non-robust training is performed on top of it. Furthermore, we find that adversarial training on top of a fixed kernel can yield a classifier with 76.1% robust accuracy under PGD attacks with $\varepsilon = 4/255$ on CIFAR-10.[1]

## 1   Introduction

Modern deep learning, while effective in tackling clean and curated datasets, is often very brittle to domain and distribution shifts [76]. Perhaps the most notorious failure mode of deep learning to domain shifts is adversarial examples [73]: images with small, bounded perturbations which consistently fool state-of-the-art classifiers. While work has been dedicated to mitigating [25, 18, 51], explaining [24, 39, 37], and harnessing [66, 54, 67] this peculiar behavior, the problem still remains largely open, with the state-of-the-art robust classifiers still falling far behind in benign accuracy compared to standard non-robust networks [18]. Indeed, tackling this idiosyncrasy of deep learning seems almost impossible given that the behavior of deep models under benign data and training is still largely unexplained [13, 62, 63], let alone adversarial training.

The community has tried developed numerous theories on better interpreting [77, 27, 43] and understanding deep learning [74, 5, 38, 64, 28, 14, 31, 30]. One emerging theory is the notion of deep learning as kernel learners [48, 21, 1]. That is, deep networks trained with gradient descent learn a kernel whose feature map is embedded in the tangent space of the network outputs with

---

[1]Code is available at `https://github.com/yolky/adversarial_ntk_evolution`

36th Conference on Neural Information Processing Systems (NeurIPS 2022).

respect to its parameters. After a brief phase of kernel learning, neural networks behave similarly to *lazy learners*, i.e., linear and slow in this neural tangent feature map. This two-stage theory of deep learning has been observed in vision networks such as residual networks [32] used in practice and has been the subject of many recent empirical [71, 48, 21, 7] and theoretical works [29, 35, 2]. While these reports have verified this theory for benign training, no work has looked at this property at the intersection of adversarial training. To this end, here, we perform the first empirical study of how the empirical/finite neural tangent kernel evolves over the course of adversarial training.[2] In section 2, we present our experimental setup designed to isolate the effect of adversarial training on the two stage of deep learning under the kernel learner theory: 1) kernel learning and 2) linear fitting. By using this framework, we study and show the following:

1. Similar to standard training, adversarial training results in a distinct kernel which quickly converges in the first few epochs of training (section 3)

2. **Adversarial robustness can be inherited from the kernel made from adversarial training**, even when the second stage classifier has **no access to adversarial examples** during training (section 4).

3. Adversarial training is effective on top of the learned kernels given by standard training (in the frozen feature regime), providing a testbed where many of the fixed-feature assumptions present in theoretical works on adversarial training are met, while still providing high practical performance (section 7).

4. Eigenvectors in the learned NTK of networks with adversarial training contained visually interpretable features, while the initial NTK and benign training NTK do not (section 8).

## 1.1   Background and Related Works

**Neural Tangent Kernel.** Explaining the empirical success of deep learning models from a theoretical standpoint is an exciting and active line of research that is still in its infancy. One of the most popular tools for understanding neural network behavior is the neural tangent kernel (NTK) [38, 6, 4]. Under this theory training deep networks with gradient descent in the infinite-width limit corresponds to training training a kernel-based classifier, with the corresponding kernel being the NTK defined as $k_{NTK}(x, x') = \mathbb{E}_{\theta \sim p(\theta)}[\nabla_\theta f(x)^T \nabla_\theta f(x')]$, for parameter distribution $p(\theta)$ and network architecture $f_\theta$. Analogously, training an infinite width Bayesian neural network corresponds to Gaussian process regression using the NNGP kernel [55, 23, 45, 52, 56, 50]. In both these infinite-width limits, the underlying feature space is fixed, determined entirely by the architecture and parameter distribution.

The frozen nature of the kernel stands in contradiction to the ability of deep models to learn useful features and representations. Indeed, it has been shown theoretically, under different limiting assumptions, or with finite-widths, that feature learning does occur, with a time-evolving and data-dependent kernel, $k_t(x, x') = \nabla_{\theta_t} f_{\theta_t}(x)^T \nabla_{\theta_t} f_{\theta_t}(x')$, with time-dependent network parameters $\theta_t$ [78, 29, 35, 2, 58, 59]. This learned NTK aligns itself with dataset labels to accelerate training and improve generalization. Empirical evidence supports this, with works finding that this meta-kernel evolves quickly within the early stages of network training before stabilizing [48, 21, 7, 71, 40, 8]. In this new setting, there exists two stages in deep network training: 1) *kernel learning* and 2) *linear fitting*. In the first stage, the kernel rapidly evolves to align with the dataset's features and labels, while in the second phase, the kernel changes only minimally and the network behaves linearly in the NTK feature map $\phi(x) = \nabla_{\theta'} f_{\theta'}(x)\big|_{\theta'=\theta_t}$, a regime sometimes referred to as *lazy training* [17, 46].

**Adversarial Examples and adversarial training.** Adversarial examples present one of the most infamous failure modes of deep learning [73]. These are images within bounded perturbations of naturally occurring images that consistently fool deep networks over a wide array of architectures [53]. There is much literature dedicated to designing techniques for securing networks against these attacks[25, 51, 18, 26, 44, 80, 61, 16, 42]. In this work, we focus on adversarial training with iterated projected gradient descent (PGD), which seeks to minimize $\mathcal{L}_{rob} = \mathbb{E}_{x,y \sim p(x,y)}\left[\max_{x' \in B_\epsilon(x)} \mathcal{L}_{standard}(x', y)\right]$. That is, minimizing the worst-case loss of samples

---

[2]Throughout this study, we refer to the *finite width* NTK as the NTK unless otherwise specified

in a small neighborhood around training examples, where the inner maximization is computed during training using iterated PGD [25, 51].

From a theoretical standpoint, much work has been devoted to explaining the causes of adversarial vulnerability and the limitations of adversarial robustness. Popular theories include the notion of there existing "robust and non-robust features" present in data [37, 76], adversarial examples being an outcome of high dimensional geometry [24, 39, 69], an outcome of stochastic gradient descent (SGD) [70, 3], and many more [14]. Based on the NTK theory, there have been works studying the presence of adversarial examples under the NTK and NNGP kernels [22, 9, 15, 60, 10]. Recently, NTK theory has been used to generate attack methods for deep networks [75, 79]. Adversarial examples have been shown to arise in simplified linear classification settings [68, 37], which readily transfer to the NTK settings as the NTK classifier is linear in its underlying feature representation. However, these reports on the adversarial vulnerability of NTK and NNGP kernels have been focused solely on the infinite-width limit of these kernels, with no literature on the adversarial robustness of the learned and data-dependent NTK that is present in practical networks.

## 2 Experimental setup

**Definitions and problem setup.** We first define the necessary objects and terminologies for our experimental setup. Our scheme follows closely that of [21]; however, we consider the additional dimension of adversarial robustness in addition to benign accuracy. First, we define the parameter-dependent empirical NTK: $k_{ENTK,\theta}(x, x') = \nabla_\theta f_\theta(x)^T \nabla_\theta f_\theta(x')$. In the case where $\theta = \theta(t)$, where $\theta(t)$ refers the parameters of the network after training for $t$ epochs, we use the shorthand $k_t = k_{ENTK,\theta(t)}$. Next, we define three training dynamics.

1. **Standard** dynamics as $f_{\text{standard},\theta} = f_\theta(t)$, that is, network behavior with no modifications.

2. **Linearized** (also referred to as lazy) dynamics around epoch $t$: $f_{\text{lin},\theta,t}(x) = f_{\theta_t}(x) + (\theta - \theta_t)^T \nabla_{\theta'} f_{\theta'}(x)\big|_{\theta'=\theta_t}$. I.e., linearized training dynamics corresponds to performing a first-order Taylor expansion of the network parameters around a point $\theta_t$. We refer to $f_{\theta_t}$ as the parent network and $t$ as the spawn epoch. Note that in linearized dynamics, the underlying feature map $\phi(x) = \nabla_{\theta'} f_{\theta'}(x)\big|_{\theta'=\theta_t}$, and the corresponding kernel is fixed. Fort et al. [21] studied this regime and showed that in practice, deep networks training with SGD undergo two stages in training, in which the first phase was chaotic with a rapid changing kernel until some relatively early epoch $t$, and the second stage behaves like linear training about $\theta_t$.

3. **Centered linear** training (or centered for short): where we subtract the parent network's output [34, 47, 49]: $f_{\text{centered},\theta,t}(x) = (\theta - \theta_t)^T \nabla_{\theta'} f_{\theta'}(x)\big|_{\theta'=\theta_t}$. This corresponds to linear training, with the zeroth order term removed. Now the output is strictly dependent on the difference between $\theta$ and $\theta_t$, and the contribution of the first $t$ epochs is only through how it modifies the feature map. Studying this setting lets us isolate the properties of the learned kernel $k_t$, without worrying about its contribution from earlier. Efficient implementation of both these linearized versions is done using forward mode differentiation in the JAX library [11].

**Experimental design** We follow closely that of [21], where we train networks in two distance stages with an added dimension of adversarial training. Additional details are available in appendix B. *Stage 1: Standard dynamics.* We train Resnet18s on CIFAR-10 or CIFAR-100 [41] for $t$ epochs either using benign data (i.e. no data modification), or adversarial training, for $0 \le t \le 100$. *Stage 2: Linearized dynamics.* Following stage 1, we take the parent network from stage 1 and train for an additional 100 epochs with either linearized or centered linearized training dynamics. Note that for centered training, after stage 1, the networks will output entirely zeros, as the zeroth order term has no effect, so the classifier does not have a "warm start." In this stage, we train using **benign** data, with no adversarial training. Additionally, we freeze the batchnorm running mean and standard deviation parameters after stage 1, as allowing these to change would implicitly change the feature map $\phi(x) = \nabla_{\theta'} f_{\theta'}(x)\big|_{\theta'=\theta_t}$. For all experiments in the main text, we use the standard $\varepsilon = 4/255$ adversarial radius under a $L_\infty$ norm, but we verify that the results hold for $\varepsilon = 8/255$ in section 9 with additional results for $\varepsilon = 8/255$ in the appendix.

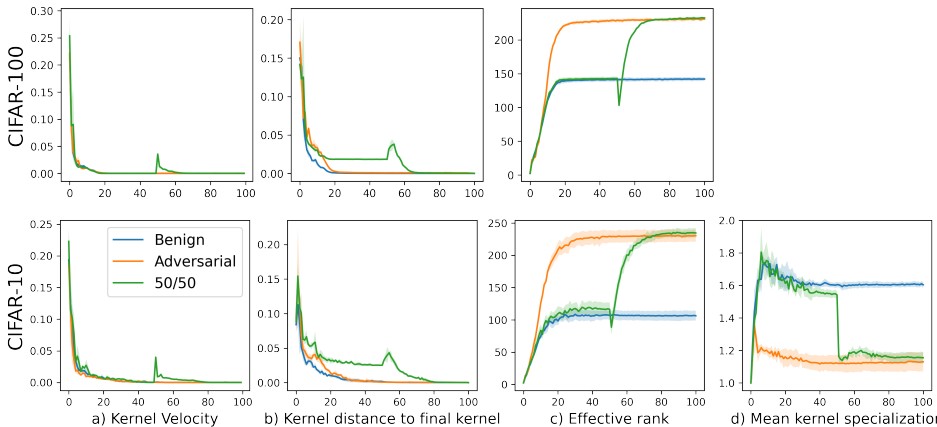

Figure 1: Evolution of the Neural Tangent Kernel under benign and adversarial training on Resnet-18s on CIFAR-100 (Top row) and CIFAR-10 (Bottom Row). Networks are either trained for 100 epochs with benign or adversarial training or 50 of benign followed by 50 epochs of adversarial training (50/50). From left to right, we plot the kernel velocity, kernel distance to the final kernel, effective rank, and mean kernel specialization of the resulting kernels. (n=3)

## 3 Evolution of the NTK under standard and adversarial training

First, we look at the evolution of the kernel under different training conditions. To do this, we calculate the empirical NTK on a random subset of 500 class-balanced training points on CIFAR-10 and CIFAR-100 for resnets trained for 100 epochs SGD under standard dynamics with either benign training or adversarial training. We also consider a third scenario where we perform benign training for 50 epochs then change to adversarial training for the remaining 50 epochs so we have an additional control point of the effect of adversarial training. For networks with multiple outputs, the kernel matrix is a rank-four tensor in $\mathbb{R}^{C \times C \times N \times N}$ with $C$ being the class count and $N$ being the dataset size. The calculation for these entries is given by $k_{c,c'}(x, x') = \nabla_\theta f_\theta^c(x)^T \nabla_\theta f_\theta^{c'}(x')$, and the resulting subclass kernel matrix $K_{c,c'} \in \mathbb{R}^{N \times N}$. Unless otherwise stated, for the statistics we present, we calculate the trace kernel, the average of the diagonal elements of the class-specific matrix: $\bar{K} = \frac{1}{C} \sum_{c=1}^{C} K_{c,c}$.

Firstly, we look at the kernel distance between neighboring epochs. This kernel distance is given by: $S(K_1, K_2) = 1 - \frac{\text{Tr}(K_1^T K_2)}{||K_1||_F ||K_2||_F}$, which equals to 0 iff $K_1 = K_2$. The kernel velocity is given by $\frac{dS}{dt}$, which we approximate as a finite difference between neighboring epochs (See fig. 1a). We also consider the kernel distance from the current epoch to the kernel at the end of training (fig. 1b). From these metrics we see that both for benign and adversarial training, the kernel converges within 30 epochs to close to the final kernel, in accordance with the results in Fort et al. [21]. This suggests that after these few epochs, the underlying feature set is fixed, and the remainder of the training is spent performing linear classification on this feature set. Surprisingly, adversarial training also quickly converges, albeit to a different kernel whose underlying feature set is more robust (as we will see in later sections). In the third setting, we observe a small spike in kernel velocity at epoch 50 when we swap from standard to adversarial training. The change is small compared to the initial rapid kernel evolution, suggesting that the standard training kernel is more similar to the adversarial kernel than it is to the initial NTK. Likewise, when we plot the kernel distance to the final kernel, both models nearly converge after epoch 40, while in the third setting the model stabilizes at the standard training kernel before changing to the adversarial kernel after 50 epochs.

The third metric we compute is the effective rank of the kernel matrix [65]. This measures the dispersion of the matrix over its eigenvectors, and is given by: $\text{erank}(K) = \exp(-\sum_{i=1}^{N} p_i \log p_i)$, With $p_i = \frac{\lambda_i}{\sum_{i=1}^{N} \lambda_i}$ with $\lambda_i$ being the eigenvalues of the kernel matrix $K$. This value is bounded between 1 when the matrix has one dominant direction and $N$ when all eigenvalues are equal. Previous work has shown that deeper networks are biased towards low effective rank (of the conjugate kernel) at initialization [36]. Alternatively, the effective rank could be interpreted as the complexity of the dataset under the kernel, and existing work has shown that robust classifiers require more

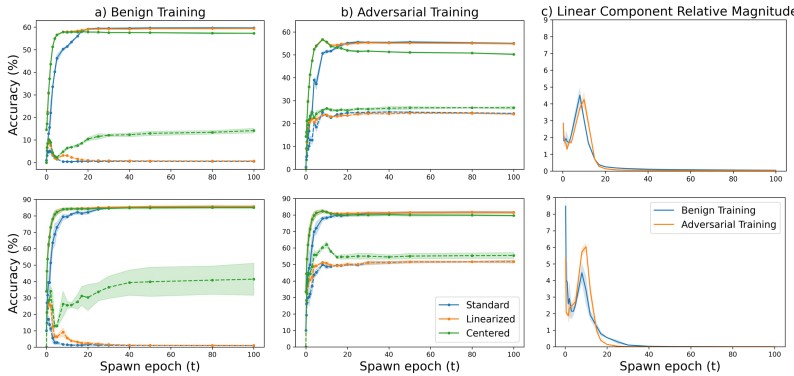

Figure 2: Performance of standard, linearized, and centered training based on kernels made from benign or standard training on CIFAR-100 (top row) and CIFAR-10 (bottom row). Solid lines indicate benign accuracy, while dashed lines indicate adversarial accuracy. Under standard or linearized dynamics with benign training (left), networks have little to no robust accuracy, but the networks learn kernels with robust features over time, as centered training gains robustness as the kernel evolves. Centered training also sees a robustness gain over adversarial training (center) at the cost of some benign accuracy. For linearized dynamics, the relative magnitude of the first-order component sharply peaks in early epochs before decaying to 0 as the spawn network full trains (right).

model capacity [14, 68], although it is unclear the precise relationship between the effective rank notion of complexity and those given in prior work relating to adversarial robustness.

As a fourth metric, we consider the mean kernel specialization. As discussed earlier, for multi-classed outputs, the full NTK is a $4D$ tensor in $\mathbb{R}^{C \times C \times N \times N}$ with C class-specific kernels corresponding to the diagonal entries of the first two dimensions. Over training, it has been observed that these individual class-specific kernels specialize in aligning themselves with their classes [71]. The kernel specialization, defined as: $\text{KSM}(c, c') = \frac{A(K^{c,c}, y_{c'} y_{c'}^T)}{C^{-1} \Sigma_{d=1}^C A(K^{d,d}, y_{c'} y_{c'}^T)}$, where $A(K^{c,c}, y_{c'} y_{c'}^T) = 1 - S(K^{c,c}, y_{c'} y_{c'}^T)$, i.e. the cosine similarity of the kernel matrix and the one-hot class labels. Intuitively, this compares how aligned class $c$ specific kernel is to the labels of class $c'$ compared to other class kernels. This quantity is bounded between 0 and $C$ and higher diagonal entries (when $c = c'$), indicate higher specialization. We define the mean kernel specialization as $C^{-1} \Sigma_{c=1}^C \text{KSM}(c, c)$, i.e. the average of diagonal entries of the KSM matrix. We calculate this for only CIFAR-10, as computing all 100 class-specific kernels for CIFAR-100 is too costly. From fig. 1d, where we see that adversarial training is associated with a lower mean kernel specialization, we take away that adversarial training promotes features that are more broadly shared between classes, although this needs further investigations which will be the focus of our continued effort.

## 4 Performance of Linearized and Centered Training

Next, we look at the performance of linearized and centered training, where we vary the spawn epoch at which we begin stage 2 training, i.e. the epoch $t$ described in section 2. We then plot the benign and adversarial performance of the classifier after stage 2 training, in comparison with the performance at the end of stage 1 training in fig. 2 for CIFAR-10 and CIFAR-100. We show the results when either stage 1 is performed with benign training or adversarial training. Additionally, for the case of linearized training, we plot the relative magnitude of the zeroth order and first-order terms in the linearization dynamics equation. Specifically, let $f_0 = f_{\theta_t}(x)$ and $\Delta f = (\theta - \theta_t)^T \nabla_{\theta'} f_{\theta'}(x)\big|_{\theta'=\theta_t}$. We plot the relative magnitude of the two components, given by $\frac{\Delta f^T \Delta f}{f_0^T f_0}$, averaged over the test set.

The most surprising observation is that centered training gains significant robustness over standard and linearized training dynamics, both in adversarial and benign training. Because centered training does include the zeroth order term in its predictions, all the robustness given by centered training is **inherited entirely through the learned NTK**, and not through adversarial training (as we perform benign training in stage 2). This lends credence to the "robust feature" hypothesis given in Ilyas et al. [37], however, now what matters is not necessarily what features are present in the data, but

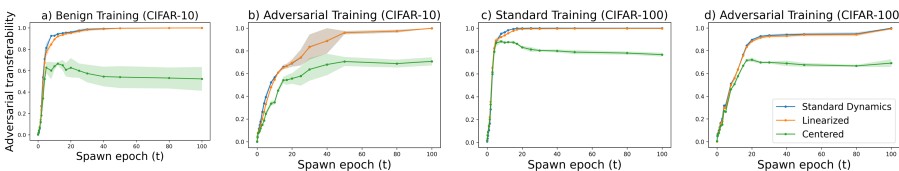

Figure 3: Adversarial transferability of standard, linearized and centered networks to fully trained networks with either benign or adversarial training on CIFAR-10 and CIFAR-100. (n=3)

what features the networks learns in its kernel. On the final adversarial kernel, centered training brings about a 3.7% adversarial accuracy gain over standard adversarial training on CIFAR-10, going from $51.77 \pm 0.80\%$ to $55.46 \pm 1.79\%$. Similar improvements are also seen on CIFAR-100 with an improvement from $24.42 \pm 0.14\%$ to $27.35 \pm 0.41$. At around epoch 10, the centered network reaches $62.09 \pm 1.76\%$ robust accuracy on CIFAR-10, an over 10% performance gain.

Equally surprising is that even under benign training, the network learns a robust kernel over time, with centered training reaching $41.41 \pm 9.75\%$ after epoch 100 for CIFAR-10 and $14.16 \pm 1.03$ for CIFAR-100. Despite this, the neural network under standard dynamics still sees no adversarial accuracy, suggesting that the contribution from the robust learned kernel is minor compared to the contribution of the non-robust kernels, which are present early in training. Roughly speaking, the final network outputs could be written as the integral over time of instantaneous kernel-dependent functions [19]. We can verify that the contribution of the kernel at later epochs is small by the relative magnitude plots in fig. 2c, which shows that the first-order additional term of linearized dynamics becomes very minor by epoch 30, meaning that the network has essentially fully fit the dataset, at which point the learned kernel reaches it maximum robustness (as seen by the centered plots). We observe that the learned kernel only gains robustness over time, but its robustness stagnates as kernel velocity slows down. Linearized training sees a peak in the relative magnitude of first to zeroth order components around epoch 10-15, coinciding with the small bump in robustness for the benign training kernel under linearized training, and for the adversarial training kernel, a peak in accuracy in benign and adversarial settings for linearized and centered training. We conjecture that this bump in relative magnitude correlates with the stage transition between kernel learning and linear fitting, where the kernel is well aligned with the data, but the network has not yet fully fit the training data, causing the first-order component to contain the majority of the data fit.

## 5   Adversarial Transferability of Linearized Dynamics

We next study the differences between standard and linearized dynamics from the perspective of adversarial transferability, specifically, how well do adversarial examples generated with one dynamics transfer to models with other dynamics? From the experiments described in section 2, we have a set of classifiers with standard, linearized, and centered dynamics: $\{f_{\text{stan},t}\}, \{f_{\text{lin},t}\}, \{f_{\text{cen},t}\}$ for $0 \leq t \leq 100$. For each of these classifiers, we generated adversarial examples on the test set with $L_\infty$ PGD (details provided in appendix B), $\{\hat{X}_{\text{stan},t}\}, \{\hat{X}_{\text{lin},t}\}, \{\hat{X}_{\text{cen},t}\}$. To see how these adversarial examples compare to those generated by a network with standard dynamics, we then evaluated the accuracy of $f_{\text{stan},t=100}$ on these adversarial examples: $\text{Acc}_{\text{stan},t=100}^{(\cdot),t'} = \mathbb{E}_{x,y \sim \hat{X}_{(\cdot),t'}}[\mathbb{I}(y = f_{\text{stan},t=100}(x))]$.

We then define the adversarial transferability as: $\tau_{(\cdot),t'} = \dfrac{\text{Acc}_{\text{stan},t=100}^b - \text{Acc}_{\text{stan},t=100}^{(\cdot),t'}}{\text{Acc}_{\text{stan},t=100}^b - \text{Acc}_{\text{stan},t=100}^{\text{stan},t=100}}$, where

$\text{Acc}_{\text{stan},t=100}^b$ is the accuracy of the standard dynamics classifier with a benign test set. This metric measures the relative effectiveness of adversarial examples generated by another network to adversarial examples generated by the network itself. Values of $\tau = 1$ mean that the adversarial examples perform just as well as adversarial examples generated by the classifier itself, while $\tau = 0$ would imply that adversarial examples do not transfer whatsoever.

We plot these for CIFAR-10 and CIFAR-100 trained Resnet-18s with either benign or adversarial training in fig. 3. Comparing the plots of benign training and adversarial training, we see that in both cases, the adversarial examples of the standard and linearized dynamics transfer similarly well to the full trained network. In contrast, the adversarial examples made by centered training are far less effective, suggesting that the feature set used by the kernel does not necessarily correspond to the

Table 1: Performance of stage 2 network training with a parent network trained with adversarial training for 10 epochs on CIFAR-10. $\eta$ = learning rates. (n=3)

| | | Frozen Batchnorm | | | Standard Batchnorm | | |
| | Parent Network | Centering | SGD $\eta = 0.0001$ | SGD $\eta = 0.01$ | Centering | SGD $\eta = 0.0001$ | SGD $\eta = 0.01$ |
|---|---|---|---|---|---|---|---|
| Benign Accuracy | $78.33 \pm 1.07$ | $81.73 \pm 0.76$ | $82.11 \pm 0.68$ | $82.21 \pm 0.96$ | $30.80 \pm 9.76$ | $52.46 \pm 18.47$ | $43.22 \pm 17.72$ |
| Adversarial Accuracy | $48.73 \pm 1.52$ | $62.09 \pm 1.76$ | $44.99 \pm 0.75$ | $44.39 \pm 1.23$ | $19.45 \pm 6.21$ | $20.91 \pm 8.69$ | $14.28 \pm 5.92$ |
| Kernel Distance | - | $0 \pm 0$ | $0.0140 \pm 0.0009$ | $0.0290 \pm 0.0042$ | $0.0012 \pm 0.0001$ | $0.0228 \pm 0.0061$ | $0.0624 \pm 0.0151$ |

same feature set used by a fully trained network. For standard training, the adversarial transferability plots rise much more quickly than for adversarial training, suggesting that features from early kernels are used more in the final model than for adversarial training, as seen by the slower initial rise of the standard and linearized training in fig. 3b and fig. 3d. As seen earlier, centered training on earlier NTK kernels provides little robust accuracy, suggesting that some of the lack of robustness of fully trained networks could be due to the contribution of these early kernels. For standard training, the adversarial transferability of fully trained networks rises sharply in the first 15 epochs, then slowly decays over time. This suggests that the adversarial behavior of the fully trained network is closer to some of the earlier kernels than to the final kernel. However, in both cases, adversarial examples generated by the initial NTK do not transfer at all to fully trained networks, suggesting that the initialization NTK analysis of adversarial examples in some theoretical work [22, 9] may not be accurately describe the behavior of adversarial examples in practical networks.

## 6 SGD vs Linearized Dynamics at low learning rates

In section 4, we showed that adversarial robustness could arise from linearized and centered training, in which the underlying feature set is fixed. In this section, we investigate whether freezing the kernel is necessary for robustness gains. To investigate this, we trained networks with adversarial training with standard dynamics until epoch 10. We then performed centered training on top of the resulting kernel. We chose ten epochs as it corresponds to the large jump in robust accuracy in fig. 2. The centered network serves as the control when the feature set is frozen. Then, we varied stage 2 training in two ways. First, we trained a network using "centered standard" dynamics, where $f_{\text{cen, stan},\theta}(x) = f_\theta(x) - f_{\theta_t}(x)$, that is, we subtract the network from its initialization's output, and train with standard gradient descent. We use learning rates of either $\eta = 0.0001$ or $\eta = 0.01$, corresponding to low and high learning rates. Previously, we had argued that freezing the batchnorm running mean and standard deviation is necessary so that the feature map $\nabla_{\theta'} f_{\theta'}(x)\big|_{\theta'=\theta_t}$ does not change. Here, we question this assumption and either keep the batchnorm parameters frozen or let them change.

Table 1 shows the resulting benign and adversarial accuracies on CIFAR-10, as well as the kernel distance between the final network's kernel and the kernel after stage 1. Additional results for different spawn epochs and for CIFAR-100 are reported in appendix C. We see that in the frozen batchnorm setting, both SGD variants lose robust accuracy. SGD does not freeze the kernel, we also see that the kernel changes too (as seen from the kernel distance), with SGD with a larger learning rate seeing a larger change. Letting the batchnorm parameters change results in low adversarial and benign accuracy for all training models, with correspondingly higher kernel distances. This interpretation of freezing batchnorm parameters to preserve the NTK provides a different perspective as to why more sophisticated batchnorm schemes improve performance in few-shot learning [12].

## 7 Adversarial Training on a Frozen NTK

In fig. 1, we observed that adversarial robustness could be inherited from a kernel in stage 1 of training, even if stage 2 was performed on benign data. We now investigate whether we can gain robustness by performing adversarial training in stage 2 on top of a frozen kernel. Much theoretical analysis of adversarial robustness has been in the frozen kernel regime or linear classification [22, 9, 60, 37, 68], as analysis of neural network training simplifies when the underlying feature set does not change.

For this experiment, we ran adversarial training using centered dynamics for three kernels. First, we used the initialization kernel, that is, $K_{t=0}$. Then we used the post-training kernels for networks

Table 2: Centered network performance with benign or adversarial training in stage 2. (n=3). The left column refers to the base kernel used to perform centered dynamics training with. The top row refers to whether benign or adversarial training was done using this kernel. The second row refers to whether adversarial attacks were used during testing.

| Base Kernel | Benign | | Adversarial | |
|---|---|---|---|---|
| | Benign Accuracy | Adversarial Accuracy | Benign Accuracy | Adversarial Accuracy |
| Standard Adversarial Training (SGD, No Kernel) | $81.58 \pm 0.63$ | $51.77 \pm 0.80$ | - | - |
| Initialization Kernel $K_{t=0}$ | $34.11 \pm 1.35$ | $0.00 \pm 0.00$ | $10.46 \pm 0.66$ | $6.27 \pm 4.45$ |
| Benign Training $K_{t=100,\text{benign}}$ | $85.06 \pm 0.75$ | $41.41 \pm 9.75$ | $84.85 \pm 0.65$ | $\mathbf{76.15 \pm 1.91}$ |
| Adversarial Training $K_{t=100,\text{adv}}$ | $79.65 \pm 0.30$ | $55.46 \pm 1.79$ | $81.23 \pm 0.57$ | $57.50 \pm 1.56$ |

Figure 4: Visualizations of the top 4 eigenvectors of the initial NTK, benign training NTK, and adversarial training NTK. We either maximize the cosine similarity (top row) or minimize it (bottom).

trained either using benign training or adversarial. As a baseline, we compare the accuracies achieved by adversarial training under standard dynamics. These results are shown in table 2, with additional results on CIFAR-100 in appendix D.

The initialization kernel fails to learn the dataset under adversarial training, achieving near-random benign accuracy and no meaningful adversarial accuracy. Adversarial training sees a very limited gain in robust accuracy, improving only $2\%$, and most surprisingly, the benign training kernel sees a dramatic robustness gain, achieving $76.15\%$, surpassing the robustness of all other methods described in this paper. Similar results are reported in appendix D for CIFAR-100, with the adversarial kernel seeing little gain and the benign kernel surpassing the robustness of all other methods. While we do not have a clear explanation as to why we see such a dramatic improvement, we do have a few speculative hypotheses. Firstly, it has been argued that fitting global structure before fitting local structure can improve robust accuracy [16, 61]. Under this view, we can consider stage 1 training as learning global structure by choosing task predictive features and stage 2 as fitting local structure by performing adversarial training. Secondly, separating the two tasks can reduce some of the instability caused by adversarial training [26, 72], as only the linear weights changed in stage 2.

From a theoretical perspective, adversarial training in linear classification is far easier to analyze, and thus this regime has been the interest of several papers [68, 22, 9]. However, in practice, these analyses have not been very practically useful as neural network behavior needs to deviate far from the linear ideal in order to achieve high standard accuracy [47]. As we have shown here, linearized training can achieve high benign and robust accuracy, provided that we use the learned NTK as opposed to the initial one. This linearized regime can thus serve as a middle ground where many of the analysis tools of adversarial training are applicable while still retaining high practical performance.

## 8    Visualizing NTK eigenvectors

In the lazy dynamics regime, the resulting classifier is linear in its features given by $\phi(x) = \nabla_{\theta'} f_{\theta'}(x)\big|_{\theta'=\theta_t}$. Furthermore, the effective classifier weights $(\theta - \theta_t)$ lie in the span of the space of the features of the training set: $J = [\phi(x_1), \phi(x_2)...\phi(x_N)]^T \in R^{N \times P}$, with $P$ the number of parameters. The top eigenvectors of the NTK $K = JJ^T$ correspond to the dominant features of the dataset in the NTK feature space. While much work has looked at the properties of the NTK eigenvectors/value at initialization to understand neural network generalization or how the eigenspace evolves over training, here we visualize these eigenvectors directly. It has been shown that not only are adversarial examples generated by robustly trained networks more visually interpretable but also that they make reasonable generative models by directly maximizing the class logits $\log p(y = c | f_\theta(x))$

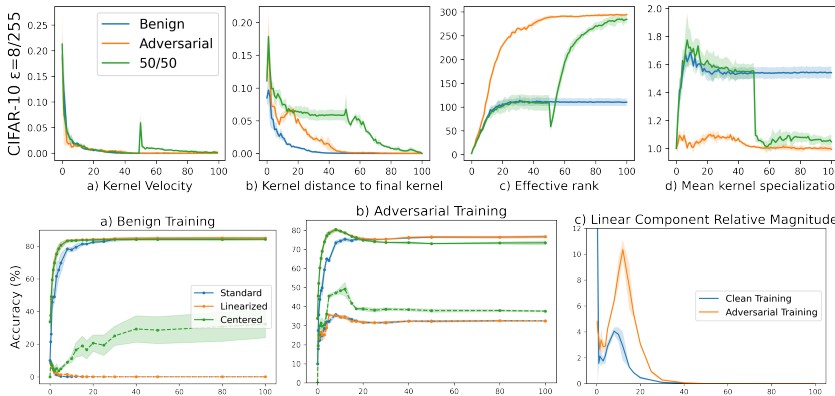

Figure 5: Evolution of the Neural Tangent Kernel under benign and adversarial training on Resnet-18s on CIFAR-10 with a larger attack radius of $\varepsilon = 8/255$ (Top row), as well as the performance of standard, linearized and centered dynamics.

w.r.t to the inputs $x$ [20]. By visualizing the eigenvectors of the network directly, we can disambiguate whether this property is caused by the linear fit on top of the eigenvectors or directly from the feature space caused by adversarial training.

To visualize these eigenvectors, we describe the following algorithm. First, we calculate the feature space representation of the top eigenvectors. This corresponds to the first columns of $V$ in the singular value decomposition of the training set features: $J = U\Sigma V^T$. Typically $P > 10^6$ and $N \sim 10^5$, so calculating this SVD directly is prohibitive. Instead, we note that $\Sigma V^T = U^T J$, so the $i$th eigenvector $v_i \propto J^T u_i$ (because $\Sigma$ is a rectangular diagonal matrix), where $u_i$ is the $i$th eigenvector of $K$. We can calculate $K = U\Lambda U^T$ since $N << P$. Calculating $J$ directly is too expensive, so instead, we note that $J^T u_i = \frac{\partial}{\partial \theta}(u_i^T f_\theta(X))$, i.e., we take derivatives of a weighted sum of the network on training samples, with respect to the parameters.

Now with the feature space representation of eigenvectors $v_i$, we visualize them by optimizing $x_i$ to optimizing $\cos(v_i, \phi(x_i))$, giving us $2N$ images, with the factor 2 arising from whether we maximize or minimizing the cosine similarity. As this algorithm describes how to visualize eigenvectors for only scalar functions $f$, we chose the dog class of the neural network as the chosen function to visualize, however, in practice, we found that eigenvectors for other classes looked very similar. As before, computing $K$ is too expensive for the full dataset, so we pick a subset of 500 images from the training set to calculate the NTK.

The visualizations of the top 4 eigenvectors of the kernels generated for the initial NTK, the kernel after 100 epochs of benign training, and the kernel after 100 epochs of adversarial training are visualized in fig. 4. The initial NTK has visually meaningless eigenvectors which look indistinguishable from pure noise. Meanwhile, the eigenvectors after benign training seem to distinguish the edges of CIFAR-10 classes but still are largely uninterpretable. Finally, the adversarial kernel results in the eigenvectors that clearly belong to different classes. This observed difference between NTK eigenvectors of the adversarial and benign NTKs provides support to the "feature purification" mechanism described in Allen-Zhu and Li [3], in which they show that adversarially trained networks provably learn robust features.

## 9 Higher $\varepsilon$ regime

In the previous sections, we showed various properties of benign and adversarial finite NTKs under the $\varepsilon = 4/255$ adversarial budget. However, it is known that the behavior of adversarial training can vary greatly depending on $\varepsilon$. In this section, we verify that the results we observed for $\varepsilon = 4/255$ hold for $\varepsilon = 8/255$. We repeat all experiments with $\varepsilon = 8/255$. fig. 5 shows the resulting kernel evolution plots for CIFAR-10, and the resulting performance of standard, linearized, and centered training dynamics.

Consistent with our previous observations, we notice that centered dynamics see an increase of robust accuracy over standard dynamics ($49.15 \pm 3.49\%$ vs. $35.80 \pm 0.76\%$) and that $32.29 \pm 8.36$ robust accuracy can be achieved from centered training on the benign kernel. We observe the robust

overfitting phenomenon, where the best adversarial accuracy occurs earlier in training. Comparing fig. 1 to fig. 5, we observe that there is a slightly larger kernel distance between the benign kernel and the adversarial one with larger $\varepsilon$ values and also that the adversarial kernel takes longer to converge with larger $\varepsilon$. This motivates future work to look at how different $\varepsilon$ values affect the stability and convergence of the finite NTK. Additionally, in appendix C and appendix D, we show that the results in section 6 and section 7 also hold for $\varepsilon = 8/255$.

## 10 Discussions, limitations and Conclusions

The experiments in this paper showed that adversarial training results in a distinct and measurable change in the NTK compared to standard training. While under both training regimes, the NTK converges quickly in the first few epochs of training, the adversarial NTK appears to have a high effective rank and lower mean kernel specialization compared to the standard training NTK. Visually, the top eigenvectors in the adversarial NTK correspond to much more visually interpretable images compared to the standard NTK and the initialization NTK.

Furthermore, we showed that classifiers built on top of the adversarial Kernel retain significant amounts of adversarial accuracy, despite being trained without ever seeing adversarial examples. The initial NTK results in classifiers with no adversarial robustness, and surprisingly, the standard training NTK sees a dramatic increase in adversarial robustness over time. We additionally demonstrated that a frozen feature set is necessary for retaining the full robustness aspect of these kernels, as either allowing the batchnorm parameters to vary or using SGD as opposed to linearized training in the second stage results in a drop in robust accuracy.

Our results strongly motivate further research utilizing the adversarial properties of linearized training we observed as a defense mechanism for adversarial attacks. Furthermore, we note that linearized training costs approximately double the cost of benign training, however, adversarial training is significantly more expensive. As shown in fig. 2, one can perform "kernel early-stopping," where we switch to linearized training on clean data so that the remainder of training can be done without adversarial training. This observation certainly motivates a more efficient way to gain adversarial robustness in practice, which future work can investigate further.

**Limitations.** We performed our analysis on robustness to an $L_\infty$ PGD adversary, which is an uncertified attack but was designed to fool networks under standard dynamics and is unproven against networks under centered or linearized dynamics. More advanced state-of-the-art adversarial defense mechanisms were not studied in this paper, nor was the optimal tuning for hyperparameters for adversarial or linearized training. Furthermore, while we presented many interesting empirical insights about adversarial training on kernels, there is a great opportunity for developing the theory of the observed interplay of adversarial training and NTK evolution.

Previous research had to impose limiting assumptions, e.g., fixed kernels on NTKs due to the nonlinear nature of neural nets [68, 22, 9]. We hope that our study of adversarial training in the frozen kernel regime with learned kernels could bridge the gap between these two extremes by having simple linear dynamics but still matching or exceeding the performance of standard networks. As shown in section 7, the performance of adversarial training in the frozen feature regime can significantly exceed that of standard dynamics. Moreover, based on our observations, we believe that centered and linearized dynamics can be used as a defense against adversarial examples.

## Acknowledgments

This research has been funded in part by the Office of Naval Research Grant Number Grant N00014-18-1-2830, DSTA Singapore, and the J. P. Morgan AI Research program.

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
