# OpenReview forum: "Evolution of Neural Tangent Kernels under Benign and Adversarial Training"
_NeurIPS.cc/2022/Conference — NeurIPS 2022 Accept_

### Official Review · Reviewer_47zX · 2022-07-07

**Rating:** 6
**Confidence:** 4
**Soundness:** 3 good
**Presentation:** 3 good
**Contribution:** 3 good

**Summary:**

This paper conducts empirical studies towards neural tangent kernel (NTK) in adversarial training. They figure out that NTK converges to a different kernel in adversarial training compared to clean training, and the new kernel provides adversarial robustness even when further fine-tuning on non-robust training.

**Questions:**

Besides the weaknesses part, I'm also wondering, is there any analytical formula to describe where the adversarially trained NTK converges (even for simple models)?

**Strengths And Weaknesses:**

Strengths:

[1] This paper provide many experiments studying the properties of adversarially trained NTK.

Weaknesses:

[1] Some paragraphs need clarifications. For example, in abstract, while most parts describing the robustness of the adversarially trained TNK, the last sentence "we find that adversarial training on top of a fixed kernel.." suddenly switch another context. Since there are a lot of things contained in this paper, please make necessary connections among different contexts.

[2] An important missing literature:

Allen-Zhu, Zeyuan, and Yuanzhi Li. "Feature purification: How adversarial training performs robust deep learning." 2021 IEEE 62nd Annual Symposium on Foundations of Computer Science (FOCS). IEEE, 2022.

Allen-Zhu et. al. (2022) provides theoretical justification on why adversarial training help purify the features. This is quite related to the 4th contribution mentioned in the submission. Please cite this paper and make comparisons or connections.

[3] It would be great if the authors could provide more insights on how people could utilize the properties of adversarially trained NTK in real practice. This paper observes interesting phenomenons for potential theoretical studies, but it could be even better if people can directly utilize the results.

---

> ### Author Response · Authors · 2022-08-02
> **Reviewer 47zX response**
>
> We would like to thank the reviewer for evaluating our work. Before going to our point by point response to review, we would like to refer the reviewer to the evaluation of our work by Reviewer 7GsC, who states with great detail the impact and importance of our presented results.
> Here we address the points raised by the reviewer:
>
> **[1]** Thanks for denoting this. We will make sure to clarify the abstract in the best way possible to connect different contexts.
>
> **[2]** Thank you for denoting the missing reference. We will add citations to it in our revised manuscript where appropriate.
>
> **[3]**  Our results strongly motivate further research utilizing the adversarial properties of linearized training we observed as a defense mechanism for adversarial attacks. Furthermore, we denote that linearized training costs approximately double the cost of benign training, however adversarial training is significantly more expensive. As we see in Figure 2, one can perform “kernel early-stopping” where we switch to linearized training on clean data, so that the remainder of training can be done without adversarial training. This observation certainly motivates a more efficient way to gain adversarial robustness in practice, which will be the focus of our continued effort.
>
> **Questions**
>
> “Besides the weaknesses part, I'm also wondering, is there any analytical formula to describe where the adversarially trained NTK converges (even for simple models)?”
>
> Currently, we do not have a way to analytically describe where the adversarial NTK converges. Note that the convergence of the standard NTK is still very difficult to describe for simple models, and the analytic formula for even the simplest models is very complex [1].
>
> [1] Huang, Jiaoyang, and Horng-Tzer Yau. Dynamics of Deep Neural Networks and Neural Tangent Hierarchy. Sept. 2019. arxiv.org, https://doi.org/10.48550/arXiv.1909.08156.

---

> > ### Comment · Reviewer_47zX · 2022-08-04
> > **Thanks for your reponse**
> >
> > I appreciate the authors addressing my concerns. It would be great if the authors could add a remark or mention in the conclusion section about [3]. I have raised my score from 5 to 6.

---

> > > ### Author Response · Authors · 2022-08-06
> > > **Thank you very much**
> > >
> > > We would like to thank the reviewer very much for increasing their review score and advocating for the acceptance of our paper.
> > >
> > > Moreover, we are so glad that our response clarified their concerns.
> > >
> > > We will certainly discuss [3] critically in the conclusions section of the revised version as instructed by the reviewer.
> > >
> > > Thank you once again,
> > > Authors

---

### Official Review · Reviewer_7GsC · 2022-07-09

**Rating:** 8
**Confidence:** 5
**Soundness:** 3 good
**Presentation:** 3 good
**Contribution:** 4 excellent

**Summary:**

This paper summarizes a novel empirical study on the adversarial robustness of neural networks using a neural tangent kernel perspective comparing the dynamics of linearized neural networks and their non-linear counterparts. The main new insights of this work are:

1. The qualitative dynamics of the NTK during adversarial training are very similar as the ones of standard training, albeit the NTKs converge to very different kernels.
2. Remarkably, even during standard training, the NTK learns some robust features that can be exploited during linearized fine-tuning.
3. Linearized and centered fine-tuning of a neural network, both adversarially and standardly pretrained, can give an edge in robustness over non-linear finetuning. This does not happen for uncentered fine-tuning.
4. Adversarial attacks transfer between linearized networks and non-linear networks mostly for the uncentered networks, and after a few epochs of training.
5. Certain visualizations of the NTK eigenfunctions of an adversarially pretrained network seem semantically more meaningful than the NTK eigenfunctions of a standardly pretrained one.


**Questions:**

I would encourage the authors to comment on the two outlined weaknesses before by providing results on other $\epsilon$ regimes (mostly on $\epsilon=8$) and clarifying the reasons for the lower benign accuracy of the centered kernel in Table 2. If the authors did this, I would be quite inclined to increase my score.

**Limitations:**

I see no direct negative societal impact stemming from this work.


**Strengths And Weaknesses:**

## Strenghts

1. **Timely and interesting perspective**: The NTK perspective has been the object of intense scrutiny by the generalization community in the past few years. In this regard, this work nicely complements these prior studies by filling a gap in the literature analyzing through the NTK lense different phenomena related to adversarial robustness. This will not only be useful for the robustness community, but also for the deep learning theory community at large, since studying adversarial training through this lense can give complementary information of the general dynamics of deep learning, as this is a very different algorithm.

2. **Remarkable findings**: Personally, I find quite remarkable that benign linearized finetuning of a standardly pretrained NTK can yield some robustness. Also, the fact that you can gain a performance edge by centered and linearized fine-tuning of an adversarially pretrained network is fascinating. This is very significant as it provides very clear new evidence that sometimes the so-called "rich regime" can have a detrimental effect in performance as shown in (Ortiz-Jimenez et al. 2021).

3. **Balanced and self-contained analysis**: In general, I found this paper very easy to read and quite balanced. The authors touch upon different topics with the right amount of breadth and depth that gives the reader a good amount of new information.

4. **Opens new research avenues**: As a new empirical study on relevant deep learning phenomena, I significantly appreciate that this work gives a clear path forward for future theoretical and practical studies. In particular, I believe that the analysis of the after-kernel of adversarially trained networks could yield significant insights for robustness, while the study of the benefitial edge from linearized fine-tuning will be of broad interest to the deep learning theory community.

## Weaknesses

1. **Limited evaluation of different $\epsilon$ regimes**: I honestly believe that this work could easily become much stronger if it provided some additional analysis of the effect of the adversarial budget $\epsilon$ on its findings. It is widely known in the adversarial robustness community that the properties and dynamics of adversarially trained networks for different $\epsilon$ can be qualitatively very different. In this regard, I believe that showing the same results for other $\epsilon$, and in particular for the standard $\epsilon=8$ regime would be very valuable. For example, if the claimed edge of linearized fine-tuning is true in this regime as well (given the right evaluation), it would suggest a radically new way to improve the robustness of neural networks.
2. **Suspicious benign accuracy of centered NTK in Table 2.**: It is a bit concerning that the benign accuracy of the centered NTK in Table 2 is lower than the robust accuracy. If this is true, this would point out to some weird geometry of the loss landsape wrt input, akin to the one happening in catastrophic overfitting, which would be very surprising in my humble opinion. Without a further investigation on that front, I would say it is more likely a sign of a technical flaw in the evaluation (which could be very damaging for this work) or a typo.
3. **Missing citations**: I have spotted a few missing citations of relevant papers that also describe the evolution of the NTK and compare it to the linearize dynamics. They should also be cited.

- G. Ortiz-Jimenez, S. Moosavi-Dezfooli, P. Frossard. "What can linearized neural networks actually say about generalization?" NeurIPS 2021.
- J. Paccolat, L. Petrini, M. Geiger, K. Tyloo, and M. Wyart, “Geometric compression of invariant manifolds in neural nets,” Journal of Statistical Mechanics: Theory and Experiment, no. 4, 2021
- A. Baratin, T. George, C. Laurent, R. D. Hjelm, G. Lajoie, P. Vincent, and S. Lacoste-Julien,“Implicit regularization via neural feature alignment,” AISTATS 2021.
- D. Kopitkov and V. Indelman, “Neural spectrum alignment: Empirical study,” in International Conference on Artificial Neural Networks (ICANN), 2020.

---

> ### Author Response · Authors · 2022-08-02
> **Reviewer 7GsC response**
>
> We would like to thank the reviewer very much for their detailed, constructive and positive evaluation of our work. We enjoyed reading their comments and perspectives, and strongly believe their insights are going to significantly improve the quality of our paper.
>
> Furthermore, we greatly appreciate that the reviewer acknowledges the importance and impact of our work in the context of the growing research interest in the NTK theory of deep learning. In the following, we try our best to address their raised concerns:
>
> **Limited evaluation of different ϵ regimes:** We fully agree with the reviewer that observing similar results on different \epsilon regimes (e.g., \epsilon=8) would suggest a new way to improve the robustness of neural nets. As requested by the reviewer, we have run additional experiments on CIFAR-10 with \epsilon=8. The figure 2 results for this experiment can be found in the following anonymized figure: https://ibb.co/Bcq50BG .
>
> As we see in the Figure, our main observations hold in this regime as well. In particular, we see that the improvement of linearized/centering fine-tuning is true in the \epsilon=8 regime as well. We are currently running the same experiment for CIFAR-100 with ϵ=8 and the remaining CIFAR-10 ϵ=8 experiments, but they will not be complete before the rebuttal period, but will certainly be included in our revised version.
>
> **Suspicious benign accuracy in table 2:** We believe the reviewer is referring to **table 1**. Indeed we incorrectly entered results into that table, and we would like to thank the reviewer very much for spotting the mistake. We also noticed the typo after the submission deadline. It is now corrected and the table is available in appendix A in the supplementary material. Note that the conclusions drawn from the text are based on the **correct** results in appendix A. Of course this correction will make its way into the main paper. We present the corrected table here for the reviewer's convenience.
>
> |                      |                    |                    | Frozen Batchnorm    |                     |                     | Standard Batchnorm  |                     |   |
> |:--------------------:|:------------------:|:------------------:|:-------------------:|:-------------------:|:-------------------:|:-------------------:|:-------------------:|:---:|
> |                      | Parent Network     | Centering          | SGD 0.0001          | SGD 0.01            | Centering           | SGD 0.0001          | SGD 0.01            |   |
> | Benign Accuracy      | $  78.33 \pm 1.07$ | $  81.73 \pm 0.76$ | $  82.11 \pm 0.68$  | $  82.21 \pm 0.96$  | $  30.80 \pm 9.76$  | $  52.46 \pm 18.47$ | $  43.22 \pm 17.72$ |   |
> | Adversarial Accuracy | $  48.73 \pm 1.52$ | $  62.09 \pm 1.76$ | $  44.99 \pm 0.75$  | $  44.39 \pm 1.23$  | $  19.45 \pm 6.21$  | $  20.91 \pm 8.69$  | $  14.28 \pm 5.92$  |   |
> | Kernel Distance      | -                  | $0\pm 0$           | $0.0140 \pm 0.0009$ | $0.0290 \pm 0.0042$ | $0.0012 \pm 0.0001$ | $0.0228 \pm 0.0061$ | $0.0624 \pm 0.0151$ |   |
>
>
>
>
> **Additional clarification for table 2**: Furthermore, we acknowledge that the caption for **table 2** (which data is **correctly** entered in) is very brief, making the results slightly confusing. We provide clarification on how to read this table here. There are three aspects to this table. Firstly, the leftmost column refers to the training configuration in stage 1 of training. Then, the upper row refers to the training configuration in stage two of centered training. Finally, the second row refers to the evaluation method. For example, the 79.65% accuracy in the bottom row, second column refers to a neural network which is trained with adversarial training under standard dynamics for 100 epochs, which then undergoes centered linear training with benign training, and is evaluated on benign data. We will provide this information in the Table 2 caption in the camera-ready version
>
> **Missing citations:** Thank you for suggesting these papers. We will add citations to them where appropriate.

---

> > ### Comment · Reviewer_7GsC · 2022-08-06
> > **Thank you for the clarifications**
> >
> > Thank you very much for the response and the clarifications, which have alleviated most of my concerns. I believe the addition of the experiments with other $\epsilon$ values will definitely make this a more solid publication, with promising early results on the use of linearisation techniques to improve robustness. That being said, I would like to point out that although I believe the results are promising, still the reported robustness numbers are quite low when compared to the standard literature in adversarial robustness (you can check RobustBench for that). This might be due to a suboptimal design choice of hyperparameters, which I hope later studies can solve. I would encourage the authors critically reflect on this fact should the paper be accepted.
> >
> > In any case, I still think this is a solid contribution to the literature which clearly deserves to be accepted to the conference. The paper provides a strong initial study on the connections between NTK theory and adversarial robustness with surprising observations both for the deep learning theory and robustness communities. I will increase my score to an 8 to acknowledge this.

---

> > > ### Author Response · Authors · 2022-08-06
> > > **Thank you very much**
> > >
> > > We would like to thank you very much once again for providing us with such a constructive, valuable, right to the point, and encouraging review.
> > >
> > > Indeed, your suggestion on adding new experiments with new $\epsilon$, signifies the impact of the work, much better.
> > >
> > > Thank you for pointing out RobustBench. We fully agree that careful design and choice of hyperprameters could improve performance to match state-of-the-art. We will certainly add a new paragraph to our discussions to include this as a limitation of the current work and to encourage future research to resolve this sub-optimality.
> > >
> > > Finally, we are so grateful that the reviewer acknowledges our contributions, advocates strongly for the acceptance of our paper, and engages with us during this rebuttal period to further improve our manuscript. Thank you.

---

### Official Review · Reviewer_36HK · 2022-07-11

**Rating:** 5
**Confidence:** 4
**Soundness:** 2 fair
**Presentation:** 4 excellent
**Contribution:** 2 fair

**Summary:**

This paper conducts an emprical study on the evolution of NTK under benign and adversarial training, which considers the training process into two phases, kernel learning and lazy training. They showed that adversarial training in kernel learning obtains a different kernel from standard training, which is robust even when non-robust training is used on top of it. And adv. training on top of a fixed learned kernel could have a high robust accuracy.

**Questions:**

See the Strength and Weakness.

**Limitations:**

See the Weakness.

**Strengths And Weaknesses:**

Strengths
- The paper is well written and easy to follow.
- It reveals some new observations through considering two phases in NTK. Particularly, benigh training could produce use kernels for the later robust linear training on top of it.

Weakness
- My major concern is that despite of the presented the empirical observations, it lacks of any tenative analysis over them.
- Is ther any practical algorithmic insights based on the obersvations found in this paper? I think as an empirical paper, good insights for algorithms is are important.
- Does these observations generalize beyond NTK?

---

> ### Author Response · Authors · 2022-08-02
> **Reviewer 36HK response**
>
> We would like to thank the reviewer for evaluating our work. Here we address some of the weaknesses pointed out in their reviews, and hope to motivate a more positive evaluation of our work:
>
> **Lacks of any tentative analysis.** We would like to emphasize that our work is the first work of its kind to evaluate the robustness properties of the learned NTK in both benign and adversarial training. As it stands, some of the properties of the learned kernel are surprising and counterintuitive - notably that without access to adversarial training, linearized training can obtain significant adversarial accuracy.
>
> Given the increasing interest in the NTK in research works studying deep learning’s remarkable representation learning capabilities, the observed phenomenon provides a new perspective of the NTK and adversarial training. Moreover, we emphasize that the analysis of any kind on the insights we gained from our paper would go beyond a single paper to wrap, but opens the opportunity for future work to explain our findings in the community.
>
> Finally, we would like to refer the reviewer to the points of strength that Reviewer 7GsC
> brought up about our paper which complements our answer to this point.
>
> **Practical Insights for algorithms.** Our results strongly motivate further research utilizing the adversarial properties of linearized training we observed as a defense mechanism for adversarial attacks. Furthermore, we denote that linearized training costs approximately double the cost of benign training, however adversarial training is significantly more expensive. As we see in Figure 2, one can perform “kernel early-stopping” where we switch to linearized training on clean data, so that the remainder of training can be done without adversarial training. This observation certainly motivates a more efficient way to gain adversarial robustness in practice, which will be the focus of our continued effort.
>
> **Do these observations generalize beyond NTK?** We are unsure what the reviewer means by this question. We would like to state that the NTK theory is a method of analyzing neural network behavior assuming a fixed feature map given by network gradients. This theory has been validated in numerous regimes, but is still the subject of intense scrutiny. Our paper aims to shed more light on how this theory can be used to understand adversarial training.
>
> Beyond the above points we address in our rebuttal, we hope that the reviewer views this work in the larger context of the deep learning theory community. As stated before, the NTK method of analyzing neural networks is an active and promising field of research, particularly with recent work studying the NTK’s evolution during training. Currently, there is a large gap in the literature which looks at the NTK evolution from an adversarial training perspective, which this paper aims to fill. We hope that our empirical observations bring to light interesting and currently unexplained phenomena of these learned kernels that future work could further study. We hope the reviewer can agree with us.

---

> > ### Author Response · Authors · 2022-08-08
> > **Final Reminder**
> >
> > We would like to take this last opportunity to kindly ask Reviewer 36HK whether their concerns are addressed with our response and if they have any additional concerns that they would like us to provide clarifications on?
> >
> > If their concerns are addressed, we hope that they get to agree with all other reviewers towards the acceptance of our submission.
> >
> >
> > Thank you,
> >
> > Authors

---

### Official Review · Reviewer_ecap · 2022-07-11

**Rating:** 6
**Confidence:** 2
**Soundness:** 3 good
**Presentation:** 3 good
**Contribution:** 2 fair

**Summary:**

The authors consider the evolution of Neural Tangent Kernels(NTK), characterization of neural networks in the infinite-width limit, under standard and adversarial training. The authors proposes the idea of adversarial robustness of NTK during training. Experiments on CIFAR-10 and CIFAR-100 show that NTK converges to a kernel and feature map. They have increased robustness, compared with standard training, which prevents catastrophic accuracy drop under the adversarial attacks. That property saves even after non-adversarial training.

**Questions:**

•	Did you use only one attack, PGD, during experiments?
•	Is it possible to use NTK adversarial approach in other domains, for example, text?


**Ethics Review Area:**

["I don’t know"]

**Limitations:**

The limitations are clearly described in the conclusion part of the paper. Authors points related works which partially cover uncovered study features

**Strengths And Weaknesses:**

Strengths:
•	Unusual use of NTK concept for researching adversarial robustness
•	Proposing and exploring three training dynamics, comparing their performance, training dynamics, and adversarial transferability with a list of custom metrics, considering properties of NTK. Those parts of the paper give a lot of insights about considering approaches
•	Great performance under adversarial attacks on the considered datasets

Weaknesses:
•	Authors consider only attacks, which were designed to fool standard dynamic networks
•	There is no theoretical substantiation of the obtained results
•	Numerical experiments only on CIFAR-10 and CIFAR-100

---

> ### Author Response · Authors · 2022-08-02
> **Reviewer ecap response**
>
> We would like to thank the reviewer for their positive evaluation of our work. Here, we try to address their questions and concerns:
>
> **Choice of adversary:** We used only L-inf PGD attacks in this study. We made this decision because these attacks are considered the standard across adversarial training literature. Our main objective was to focus this paper on the interesting behavior of linearized networks as opposed to the nuances of different attacks. Please note that this paper is one of the first to look at the adversarial robustness of neural network after-kernels and how they evolve. As a result, **there are no attacks designed to fool networks under linearized/centered dynamics** in the literature currently. Future work could look at designing such attacks enabled by our results laid out here.
>
> **Lack of theoretical analysis:** Currently, there is an active line of work trying to characterize the evolution of the NTK under benign training, however these results are still very nascent. As such, hoping to theoretically analyze the behavior of the adversarial NTK remains an open research avenue which we hope that future work will be able to address.
>
> **Evaluation of limited datasets:** We chose CIFAR-10 and CIFAR-100 as they are the standard datasets for analyzing NTK dynamics [1,2, 3]. The techniques used in this paper can generalize to any other domain such as text, however currently there is limited work studying the applicability of the NTK to these domains.
>
> [1] Fort, Stanislav, et al. Deep Learning versus Kernel Learning: An Empirical Study of Loss Landscape Geometry and the Time Evolution of the Neural Tangent Kernel. Oct. 2020. arxiv.org, https://doi.org/10.48550/arXiv.2010.15110.
>
> [2] Lewkowycz, Aitor, et al. The Large Learning Rate Phase of Deep Learning: The Catapult Mechanism. Mar. 2020. arxiv.org, https://doi.org/10.48550/arXiv.2003.02218.
>
> [3] Shan, Haozhe, and Blake Bordelon. A Theory of Neural Tangent Kernel Alignment and Its Influence on Training. May 2021. arxiv.org, https://doi.org/10.48550/arXiv.2105.14301.

---

### Meta-Review · Area_Chair_VUTg · 2022-08-27

**Recommendation:** Accept
**Confidence:** Certain

**Metareview:**

The paper studies the evolution of NTK under adversarial training. The empirical studies show that the NTK of adversarial training converges to a different kernel compared to standard training.  All the reviewers agree that the empirical study is interesting and should be accepted.


Missing citations: The actual notion of the NTK the authors consider is called finite-width NTK, which is proposed in the paper "Convergence analysis of deep learning via over-parameterization".

**Award:**

No

---

### Decision · Program_Chairs · 2022-09-14

Accept